# A conserved strategy of chalcone isomerase-like protein to rectify promiscuous chalcone synthase specificity

Toshiyuki Waki[1], Ryo Mameda[1], Takuya Nakano[1], Sayumi Yamada[1], Miho Terashita[1], Keisuke Ito[1], Natsuki Tenma[1], Yanbing Li[1], Naoto Fujino[1], Kaichi Uno[1], Satoshi Yamashita[2], Yuichi Aoki[3], Konstantin Denessiouk [4], Yosuke Kawai[5], Satoko Sugawara [6], Kazuki Saito [6], Keiko Yonekura-Sakakibara [6], Yasumasa Morita[7], Atsushi Hoshino [8], Seiji Takahashi [1] & Toru Nakayama [1✉]

Land plants produce diverse flavonoids for growth, survival, and reproduction. Chalcone synthase is the first committed enzyme of the flavonoid biosynthetic pathway and catalyzes the production of 2′,4,4′,6′-tetrahydroxychalcone (THC). However, it also produces other polyketides, including *p*-coumaroyltriacetic acid lactone (CTAL), because of the derailment of the chalcone-producing pathway. This promiscuity of CHS catalysis adversely affects the efficiency of flavonoid biosynthesis, although it is also believed to have led to the evolution of stilbene synthase and *p*-coumaroyltriacetic acid synthase. In this study, we establish that chalcone isomerase-like proteins (CHILs), which are encoded by genes that are ubiquitous in land plant genomes, bind to CHS to enhance THC production and decrease CTAL formation, thereby rectifying the promiscuous CHS catalysis. This CHIL function has been confirmed in diverse land plant species, and represents a conserved strategy facilitating the efficient influx of substrates from the phenylpropanoid pathway to the flavonoid pathway.

[1] Graduate School of Engineering, Tohoku University, Aza Aoba, Aramaki, Aoba 6-6-11, Sendai, Miyagi 980-8579, Japan. [2] Graduate School of Natural Science and Technology, Kanazawa University, Kakuma, Kanazawa 920-1192, Japan. [3] Department of Integrative Genomics, Tohoku Medical Megabank Organization, Seiryo 2-1, Sendai, Miyagi 980-8573, Japan. [4] Structural Bioinformatics Laboratory, Biochemistry, Faculty of Science and Engineering, Åbo Akademi University, Turku, Finland. [5] Genome Medical Science Project, National Center for Global Health and Medicine, 1-21-1 Toyama Shinjuku-ku, Tokyo 162-8655, Japan. [6] The RIKEN Center for Sustainable Resource Science, Yokohama, Kanagawa 230-0045, Japan. [7] Experimental Farm, Faculty of Agriculture, Meijo University, Kasugai, Aichi 486-0804, Japan. [8] National Institute for Basic Biology, Okazaki, Aichi 444-8585, Japan. ✉email: toru.nakayama.e5@tohoku.ac.jp

F lavonoids are an important class of specialized metabolites, consisting of more than 6900 different structures, that are produced by land plants (Embryophytes)[1]. After plants colonized the terrestrial environment 450 million years ago, flavonoids are believed to have played indispensable roles in plant survival, which include antioxidant and UV-screening functions as well as regulatory roles mediating the oxidative stress-induced activation of signaling cascades[2]. In extant plants, flavonoids are also involved in the mechanisms underlying plant reproduction and defense against biotic and abiotic stresses[3]. Flavonoids are derived from p-coumaroyl-CoA, which is a general precursor for a vast array of phenylpropanoids with diverse functions, such as cuticle biopolymers (in all land plants), lignins (in vascular plants), lignans, and hydroxycinnamoyl esters (Fig. 1)[4]. The flavonoid biosynthetic pathway branches off from the general phenylpropanoid pathway, and chalcone synthase (CHS) is the first committed enzyme of this pathway (Fig. 1). Chalcone synthase is a plant-specific type III polyketide synthase (PKS) that catalyzes the production of 2′,4,4′,6′-tetrahydroxychalcone (THC) from one p-coumaroyl-CoA and three malonyl-CoA molecules[5]. The resulting THC subsequently undergoes a stereo-specific isomerization catalyzed by chalcone isomerase (CHI) to produce (2S)-naringenin (a flavanone), which is metabolized to other classes of flavonoids in a plant lineage-specific manner (Fig. 1)[6]. Additionally, CHS, CHI, and other flavonoid enzymes further down the pathway are reportedly organized to form a metabolon (i.e., a fragile, highly organized super-molecular enzyme complex) using P450 protein(s) as a nucleus for the enzyme association[7–11].

In many cases, enzymes involved in plant specialized metabolism coincidentally catalyze reactions other than those for which they evolved. This nature of specialized metabolic enzymes, referred to as "catalytic promiscuity"[12,13], provides an important basis for the adaptive evolution of plant specialized metabolism. Like many other enzymes involved in plant specialized metabolism, CHS is a promiscuous enzyme, which, at least in vitro, coincidentally catalyzes the formation of other polyketides, including p-coumaroyltriacetic acid lactone (CTAL), due to the derailment of the chalcone-producing pathway (Fig. 1)[5,12]. This promiscuity of CHS catalysis likely decreases the efficiency of flavonoid biosynthesis, but it is also believed to have served as the basis for the evolution of other plant-specific PKSs, including stilbene synthase (STS)[14] and p-coumaroyltriacetic acid synthase (CTAS)[15].

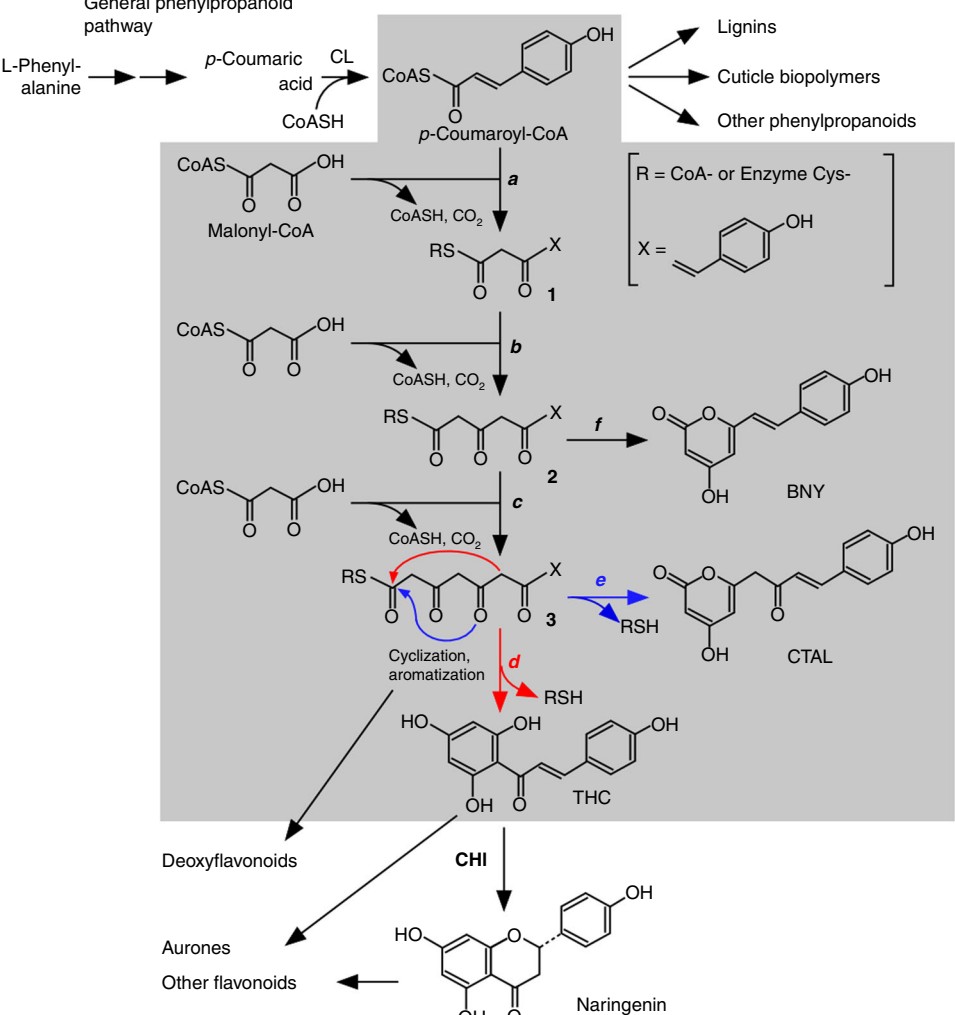

**Fig. 1 Chalcone synthase (CHS)-catalyzed formation of THC and other polyketides.** The CHS-catalyzed process (steps **a** through **f**) is presented in a gray background. One of the CHS substrates, p-coumaroyl-CoA, is derived from L-phenylalanine via the general phenylpropanoid pathway and serves as a precursor of lignin, cuticle biopolymers, and other phenylpropanoids in plants. A diffusible intermediate of the CHS-catalyzed reaction may serve as a substrate of chalcone reductase in deoxyflavonoid biosynthesis[10]. Additionally, THC serves as a precursor of aurones, whereas naringenin serves as a precursor of other flavonoids[6,11]. CL, 4-coumaric acid:CoA ligase.

In 2014, Morita et al. identified a gene in Japanese morning glory (*Ipomoea nil*) for which a loss-of-function mutation results in pale-colored flowers because of a decrease in the abundance of anthocyanins (a class of flavonoids responsible for cyan flower colors) and flavonols (a class of flavonoids involved in co-pigmentation)[16]. This gene, designated as *ENHANCER OF FLAVONOID PRODUCTION* (*EFP*), encodes a type IV CHI-like protein (CHIL) that lacks CHI activity[17]. Moreover, CHIL may have evolved from a fatty acid-binding protein (FABP) when plants colonized land[17]. Although the CHIL-mediated enhancement of flavonoid production appears to occur in many plant species[16], the underlying mechanism remains to be clarified. Recent studies indicated that CHIL interacts with CHI[18] and CHS[19] and may serve as an activator of these enzymes in *Arabidopsis thaliana*[18] and some other plant species[19].

In this study, we prove that CHIL is a component of flavonoid metabolons and binds to CHS to serve as a rectifier—rather than an activator—of promiscuous CHS; it enhances CHS-catalyzed THC production and diminishes CTAL formation without enhancing total polyketide production, thereby facilitating the efficient influx of substrates from the general phenylpropanoid pathway to the flavonoid pathway. We reveal that this role of CHIL as a rectifier for CHS is widely conserved among land plants. On the basis of the present findings, we discuss the differentiation of CHS-related enzymes (STS and CTAS) from CHS in regard to their ability to interact with CHIL. We propose that the role of CHS in THC production has been maintained throughout the evolution of land plants by the continued ability of CHS to interact with CHIL. However, some CHS homologs have lost their ability to interact with CHIL, resulting in neofunctionalizations to produce enzymes with differential product specificities, such as CTAS and STS.

## Results

**CHIL is a component of flavonoid metabolons.** To clarify the role of CHIL in the enhancement of flavonoid production, we examined the possibility that CHIL is a flavonoid metabolon component. First, we used yeast two-hybrid systems to comprehensively analyze the binary protein–protein interactions between CHIL and flavonoid biosynthetic enzymes in plants confirmed to contain flavonoid metabolons, namely snapdragon (*Antirrhinum majus* L. (Lamiales)[9]) and soybean (*Glycine max* (Fabales)[8,10]) (Supplementary Fig. 1). We observed that in both plant species, CHIL interacts with CHS and cytochromes P450 [i.e., 2-hydroxyisoflavanone synthase (IFS) in soybean and flavone synthase II (FNSII) in snapdragon] (Fig. 2). These results were confirmed in planta by means of bimolecular fluorescence complementation (BiFC) experiments (Supplementary Fig. 2), suggesting that CHIL is a flavonoid metabolon component in these plants. To further confirm this, co-precipitation experiments involving the snapdragon flavonoid metabolon (Supplementary Fig. 1a)[9] were conducted with the His$_6$-tagged snapdragon CHIL and CHS (His$_6$-AmCHIL and His$_6$-AmCHS, respectively) as bait proteins (see Methods for details). The results revealed that 150 and 192 proteins were able to specifically bind to His$_6$-AmCHIL and His$_6$-AmCHS (Supplementary Fig. 3), and AmCHS (Am04g40840.P01) and AmCHIL (Am07g21400.P01), respectively, were identified in the bound proteins. Specific binding of AmCHS to AmCHIL in the extract of snapdragon petal cells was further corroborated by the results of immunoblot analysis that followed co-precipitation (Supplementary Fig. 4). These results provided additional evidence that CHIL is a flavonoid metabolon component.

**Conservation of CHIL–CHS interactions among land plants.** Genes encoding CHIL and CHS (but not IFS and FNSII) are ubiquitous in the genomes of land plants. Thus, we analyzed the physical interactions between CHIL and CHS in phylogenetically distant land plants (bryophyte to angiosperms) (Table 1) with yeast two-hybrid systems and BiFC experiments. Consequently, CHIL and CHS were observed to interact with each other in all plant systems examined with both assay systems (Fig. 3a, c; see also Supplementary Fig. 5), strongly suggesting that this physical interaction is conserved throughout land plants. In contrast, CHI did not bind to CHS (Fig. 3a, Supplementary Fig. 5), with the exception of the enzymes from snapdragon[9] and *A. thaliana*[7]. Moreover, CHIL did not interact with the STS of grapevine (*Vitis vinifera*) (VvSTS[14]) or the CTAS of hydrangea (*Hydrangea macrophylla*) (HmCTAS[15]) (Fig. 3a, c, Supplementary Fig. 5), both of which show close phylogenetic relationships with CHS. To examine the stoichiometric characteristics of CHIL binding to CHS, a mixture containing His$_6$-tagged *A. majus* CHIL and CHS [i.e., His$_6$-AmCHIL (25 kDa, 10 μM) and His$_6$-AmCHS (monomer molecular mass[20], 45 kDa; 5 μM) was combined with bis(sulfosuccinimidyl) suberate disodium salt (BS3, a bi-functional cross linker; 4 mM) at pH 7.5 for 5 min, followed by SDS-PAGE analysis. Protein band with a molecular mass of 70 kDa was detected only for the BS3-reacted protein mixture (Supplementary Fig. 6). This band was immuno-reactive to anti-AmCHS and anti-AmCHIL antibodies and most consistently designated as an AmCHIL monomer cross-linked to an AmCHS monomer. These results were consistent with the binding of one CHIL monomer per CHS monomer. Apparent $K_D$ values of the CHIL/CHS complexes of various plant species were calculated based on biolayer interferometry (Supplementary Table 1), and ranged from 1 nM (for morning glory) to 126 nM (for gingko (*Ginkgo biloba*)). Thus, the interactions between CHIL and CHS were slightly stronger than the interactions between the proteins of many other metabolons, with $K_D$ values of 0.03–4 μM[21,22].

We then examined whether the binding of CHIL to CHS takes place in a species-specific manner. The binding of heterologous CHILs cCHILs of morning glory (InCHIL) and bryophytes (PpCHIL and MpCHIL of *Physcomitrella patens* and *Marchantia polymorpha*, respectively)) to *Arabidopsis* CHS (AtCHS) was assayed using a yeast two-hybrid system (Fig. 3b). The results showed that InCHIL bound weakly to AtCHS compared with AtCHIL, whereas PpCHIL and MpCHIL were unable to bind to AtCHS. To further confirm the species specificity of CHIL–CHS interactions in planta, *AtCHIL*, *PpCHIL*, and *MpCHIL* were respectively expressed in the *CHIL*-knockout mutant of *A. thaliana* (*chil-3*) under the control of the *AtCHIL* promoter (Fig. 4a). The seed coat of the wild-type *A. thaliana* (Col-0) was brown, whereas that of the *chil-3* mutant was paler in color primarily owing to decreased contents in the seed coat of proanthocyanidin (PA) in the mutant (Fig. 4b). The *chil-3* line expressing *AtCHIL* (*chil-3/AtCHIL*) consistently produced brown-colored seeds, which were indistinguishable from those of the wild type (Col-0) (Fig. 4b), whereas the *chil-3* line expressing *InCHIL* (*chil-3/InCHIL*) displayed a slightly paler seed-color phenotype. Seed-color phenotypes of the *chil-3* lines expressing a bryophyte *CHIL* (*chil-3/PpCHIL-A*, *chil-3/PpCHIL-B*, and *chil-3/MpCHIL*) were indistinguishable from those of the *chil-3* mutant and negative controls (*chil-3/35SGUS*, *chil-3/AtCHI*, and *chil-3/MpCHI*) (Fig. 4b). Furthermore, soluble and insoluble PAs in the seeds of the transgenic *chil-3* lines were extracted and determined spectrophotometrically. Consistently, the contents of both types of PAs in the seeds of *chil-3/AtCHIL* were fully restored to those of the wild type (Fig. 4c, d). In this connection, the *chil* mutants expressing *AtCHIL* under the control of the *Cauliflower mosaic virus* 35S promoter showed insufficient recovery of PA contents[18], suggesting that the use of the original

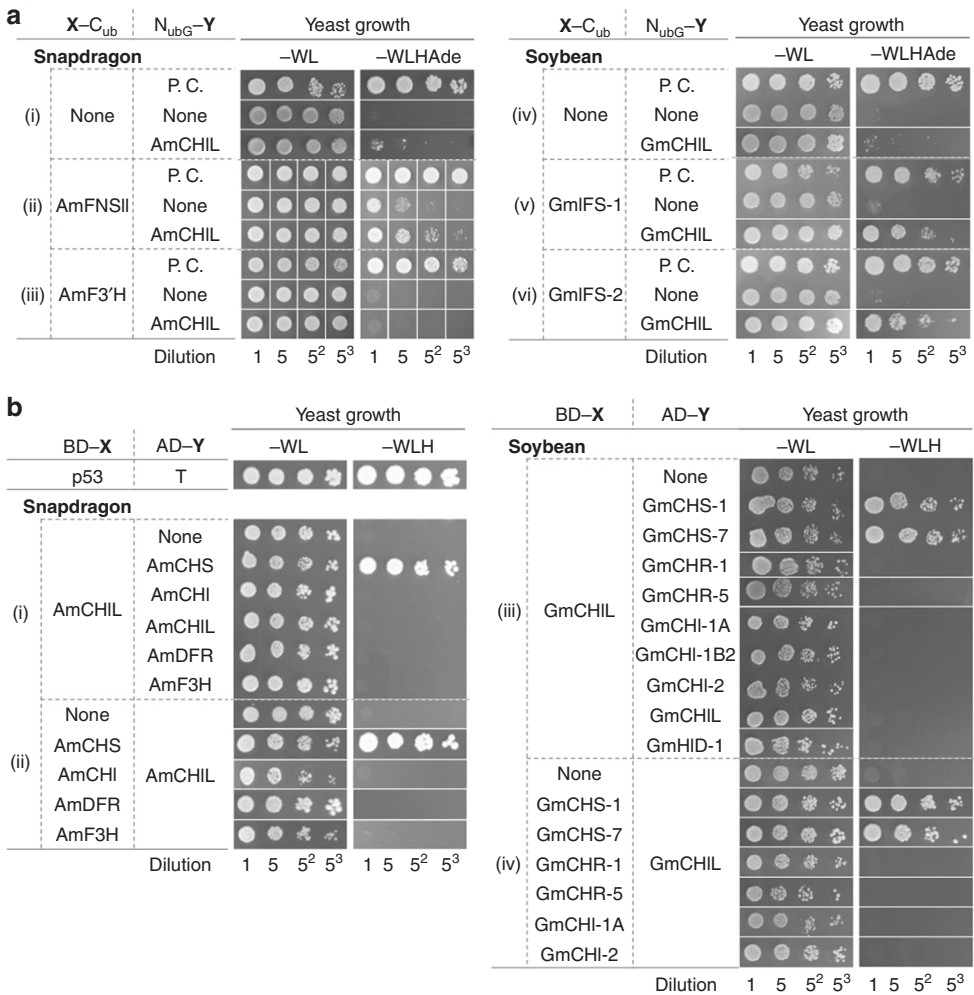

**Fig. 2 Interactions between CHIL and flavonoid enzymes in snapdragon and soybean as determined in yeast assay systems. a** Interactions between CHIL and cytochromes P450 in snapdragon (i–iii) and soybean (iv–vi) were analyzed with the split-ubiquitin yeast two-hybrid systems. Regarding the protein names, the initials of the scientific name of plant species are followed by the abbreviated enzyme/protein names, with homologs indicated with a hyphen. For example, GmCHS-1 refers to isozyme 1 of a soybean (*Glycine max*) CHS. (i–iii) Growth of yeast cells co-expressing X-$C_{ub}$-LexA-VP16 with $N_{ubG}$-AmCHIL, where X refers to (i) nothing, (ii) flavone synthase II (AmFNSII), and (iii) flavonoid 3′-hydroxylase (AmF3′H). (iv–vi) Growth of yeast cells co-expressing X-$C_{ub}$-LexA-VP16 with $N_{ubG}$-GmCHIL, where X refers to (iv) nothing, (v) isozyme 1 of 2-hydroxyisoflavanone synthase (GmIFS-1), and (vi) isozyme 2 of GmIFS (GmIFS-2). In the $N_{ubG}$-Y column of each panel, P.C. refers to the positive control [transformed cells harboring pOst1-NubI (a plasmid expressing a fusion protein comprising the yeast resident ER protein Ost1 and the wild-type $N_{ub}$ portion of yeast ubiquitin) and derivatives of pBT3-SUC]. "None" refers to the negative control (transformed cells harboring the empty pPR3-N vector and derivatives of pBT3-SUC). **b** Interactions between CHIL and soluble flavonoid enzymes in (i, ii) snapdragon and (iii, iv) soybean were analyzed with GAL4-based yeast two-hybrid systems. The growth of yeast cells co-expressing CHIL and flavonoid enzymes as fusions with the activation domain (AD, AD-Y column) and the DNA-binding domain (BD, BD-X column) of the yeast GAL4 transcription factor is presented. (i) BD-AmCHIL was co-expressed with AD-Y, where Y refers to CHS (AmCHS), CHI (AmCHI), AmCHIL, dihydroflavonol 4-reductase (AmDFR), or flavanone 3-hydroxylase (AmF3H). "None" refers to yeast cells co-expressing AD (without a fused protein) and BD-AmCHIL. (ii) AD-AmCHIL was co-expressed with BD-X, where X refers to AmCHS, AmCHI, AmDFR, or AmF3H. "None" refers to yeast cells co-expressing BD (without a fused protein) and AD-AmCHIL. (iii) BD-GmCHIL was co-expressed with AD-Y, where Y refers to CHS isozymes (GmCHS-1 and GmCHS-7), chalcone reductase (CHR) isozymes (GmCHR-1 and GmCHR-5), CHI isozymes (GmCHI-1A, GmCHI-1B2, and GmCHI-2), GmCHIL, or a 2-hydroxyisoflavanone dehydratase isozyme (GmHID-1). "None" refers to yeast cells co-expressing AD (without a fused protein) and BD-GmCHIL. (iv) AD-GmCHIL was co-expressed with BD-X, where X refers to GmCHS-1, GmCHS-7, GmCHR-1, GmCHR-5, GmCHI-1A, or GmCHI-2. "None" refers to yeast cells co-expressing BD (without a fused protein) and AD-GmCHIL. Abbreviated growth media names are as follows: −WL, SD agar medium lacking tryptophan and leucine; −WLH, −WL medium lacking histidine; −WLHAde, −WLH medium lacking adenine.

promoter may be suitable. The content of insoluble PA in the seeds of *chil-3/InCHIL* was higher than those of the *chil* mutants (*chil-1* and *chil-3*) and negative controls, although it was lower than that of the wild type (Fig. 4d). The content of soluble PA in the seeds of *chil-3/InCHIL* was only slightly higher than those of the *chil* mutants (Fig. 4c). No significant change in the contents of soluble and insoluble PAs was observed in *chil-3/PpCHIL-A*, *chil-3/ PpCHIL-B*, and *chil-3/MpCHIL*. These results suggested that the

binding of CHIL to CHS takes place in a species-specific manner and that heterologous CHIL could not fully implement its in planta function in phylogenetically distant species.

**Promiscuous CHS specificity and effect of CHIL binding.** When AmCHS was assayed with *p*-coumaroyl-CoA and malonyl-CoA as substrates under standard assay conditions (see Methods),

**Table 1 Plant sources of CHILs and flavonoid enzymes used in this study.**

| Taxonomic group | Scientific name | Common name |
|---|---|---|
| Bryophyte | *Physcomitrella patens* | Spreading earth moss |
| | *Marchantia polymorpha* | Common liverwort |
| Lycophyte | *Selaginella moellendorffii* | Selaginella |
| Gymnosperm | *Ginkgo biloba* | Ginkgo |
| Angiosperm | | |
| Monocot | | |
| Poales | *Oryza sativa* | Rice |
| Dicot | | |
| Vitales | *Vitis vinifera* | Grapevine |
| Fabales | *Glycine max* | Soybean |
| Brassicales | *Arabidopsis thaliana* | Arabidopsis |
| Cornales | *Hydrangea macrophylla* | Hydrangea |
| Solanales | *Ipomoea nil* | Morning glory |
| Lamiales | *Antirrhinum majus* | Snapdragon |
| | *Torenia hybrida* | Torenia |

two prominent product peaks (P1 and P2) and one very minor peak (P3) were observed during the LC-MS analysis (Supplementary Fig. 7a, upper panel). A subsequent tandem MS/MS analysis identified P1 and P2 as CTAL and naringenin (an isomer of THC that forms via a non-enzymatic isomerization), respectively (Supplementary Fig. 7b). The P3 peak most likely represented bis-noryangonin (BNY), according to its molecular mass and absorption spectra. Our assay results revealed that the product ratios of the CHS-catalyzed reactions varied with the enzyme source (Fig. 5a–e), with the formation of BNY being negligible in all cases (<1% (mol/mol) of all products). Thus, CHS is generally a promiscuous enzyme in terms of product specificity, irrespective of plant sources.

We then examined the effects of AmCHIL (1.0 μM) on the product specificity of AmCHS (0.1 μM) under standard assay conditions. In the presence of the added AmCHIL, the amount of THC and naringenin [collectively termed chalconoids (CLC)] formed in the reaction mixture increased 1.7-fold, with a substantial decrease in CTAL formation (Fig. 5a). The BNY-producing activity of AmCHS, which was negligible, was unaffected by the presence of CHIL. The total amount of CHS products (CLC, CTAL, and BNY) in the presence of CHIL was 116% of the amount in its absence, indicating that AmCHIL only slightly enhanced the total polyketide-producing activity of AmCHS. This was also the case for the CHS–CHIL systems of other plant species (see Fig. 5; panels b through e). To compare the product specificity of the CHS-catalyzed reaction in the presence and absence of CHIL, a CLC-excess value (i.e., E value) was defined as follows:

$$E = \frac{[\text{CLC}] - [\text{CTAL}]}{[\text{CLC}] + [\text{CTAL}]} \times 100 \, (\%)$$

where [CLC] and [CTAL] correspond to the concentrations of CLC and CTAL produced in the reaction mixture, respectively. A positive E value indicates the rate of the CHS-catalyzed production of THC exceeds that of CTAL production, whereas a negative E value indicates the rate of CTAL production exceeds that of THC production. For example, in the absence of CHIL, the E values of the reactions catalyzed by 0.1 μM AmCHS and 0.1 μM *P. patens* CHS (PpCHS, a bryophyte CHS) were +18% (Fig. 5e) and −67% (Fig. 5e), respectively. In the presence of 1 μM CHIL, the E values of the reactions catalyzed by these enzymes were +80 and +90%, respectively (Fig. 5e). Similar results were obtained for the CHS–CHIL systems of all other examined plant

species (Fig. 5e). Thus, in essence, the presence of CHIL increases the E value of the CHS-catalyzed reaction to enhance the product specificity for THC production. When a constant amount of CHS (0.1 μM) was assayed in the presence of varying amounts of CHIL, a 5- to 10-fold molar excess (on a monomer basis) of CHIL over CHS was observed to substantially alter the product specificity of CHS (Fig. 5f, g for snapdragon and *P. patens* systems). Moreover, the CHIs did not affect the E values of CHS-catalyzed reactions (Fig. 5a, c). Furthermore, CHIL did not influence the product specificities of STS and CTAS (Fig. 5c, d).

**Effects of CHIL on kinetics and in vivo specificity of CHS**. We analyzed the effects of CHIL on the kinetic parameters of the CHS-catalyzed formation of THC in the presence and absence of excess CHIL in representative plant systems. The $k_{cat}$ value of THC production was 2- to 15-times higher in the presence of CHIL than in its absence for all examined plant species (Tables 2 and 3). The $K_m$ and $K'$ values (see Methods) for *p*-coumaroyl-CoA and malonyl-CoA, respectively, were also higher in the presence of CHIL than in its absence (Tables 2 and 3).

We subsequently examined the effects of CHIL on the product specificity of in vivo chalcone synthesis in *Escherichia coli* cells. First, *E. coli* cells were genetically engineered to produce soybean 4-coumaric acid:CoA ligase (Gm4CL-3; corresponding to CL in Fig. 1) and an isozyme of soybean CHS (GmCHS-1). When the recombinant cells (strain CHIL−) were incubated with 1 mM *p*-coumaric acid at 30 °C for 3 h, the cells produced 327 nM CLC, with an E value of +66.5% (Fig. 6), which was significantly higher than the E value (−7%) obtained via in vitro enzymatic production (Fig. 5e). To clarify the effects of GmCHIL on the product specificity of in vivo chalcone synthesis, *GmCHIL* was co-expressed with *Gm4CL-3* and *GmCHS-1* in strain CHIL− (see above). The results of immunoblot analyses revealed that the molar ratio (GmCHIL: GmCHS-1, on monomer basis) of the expressed proteins in the resulting cells (strain CHIL+) was 1.5:1.0 (Supplementary Fig. 8). The CHIL+ cells were incubated with 1 mM *p*-coumaric acid at 30 °C for 3 h. The HPLC results revealed that the E value obtained with this system was +91%, indicating that the product specificity of the in vivo CHS-catalyzed reaction increased significantly (Fig. 6). The total amount of polyketides (CLC + CTAL) produced in strain CHIL+ was essentially unchanged from that obtained in strain CHIL−.

## Discussion

Enzymes involved in specialized metabolism may exhibit promiscuous catalytic properties, which are believed to serve as the basis for the functional evolution (neofunctionalization) of enzymes[12,13]. Despite the pivotal role of CHS as the first committed enzyme in the flavonoid pathway, CHS is a promiscuous enzyme irrespective of its plant origins. In vitro assay results indicated that CHS coincidentally produces a large amount of CTAL as a byproduct in addition to THC, with E values ranging from −67 to +33% depending on the plant species (Fig. 5e). The present study establishes that CHIL binds to CHS with a $K_D$ value between $10^{-9}$ and $10^{-7}$ M, thereby increasing the E value of CHS catalysis to facilitate efficient THC production (Figs. 5 and 6). A previous study demonstrated that CHIL activates the CHS-mediated production of THC[19], but we showed that it only weakly induces the PKS-catalyzed production of polyketides (THC, CTAL, and BNY). Thus, in essence, CHIL is considered to be a rectifier—rather than an activator—of promiscuous CHS catalysis. Of note, during the review of this paper, a research group independently reported the ability of a fern CHIL to alter product specificity of CHS[23], consistent with our results. The *CHIL* genes are ubiquitous in the genomes of land plants[17] and expressed along with *CHS* genes in

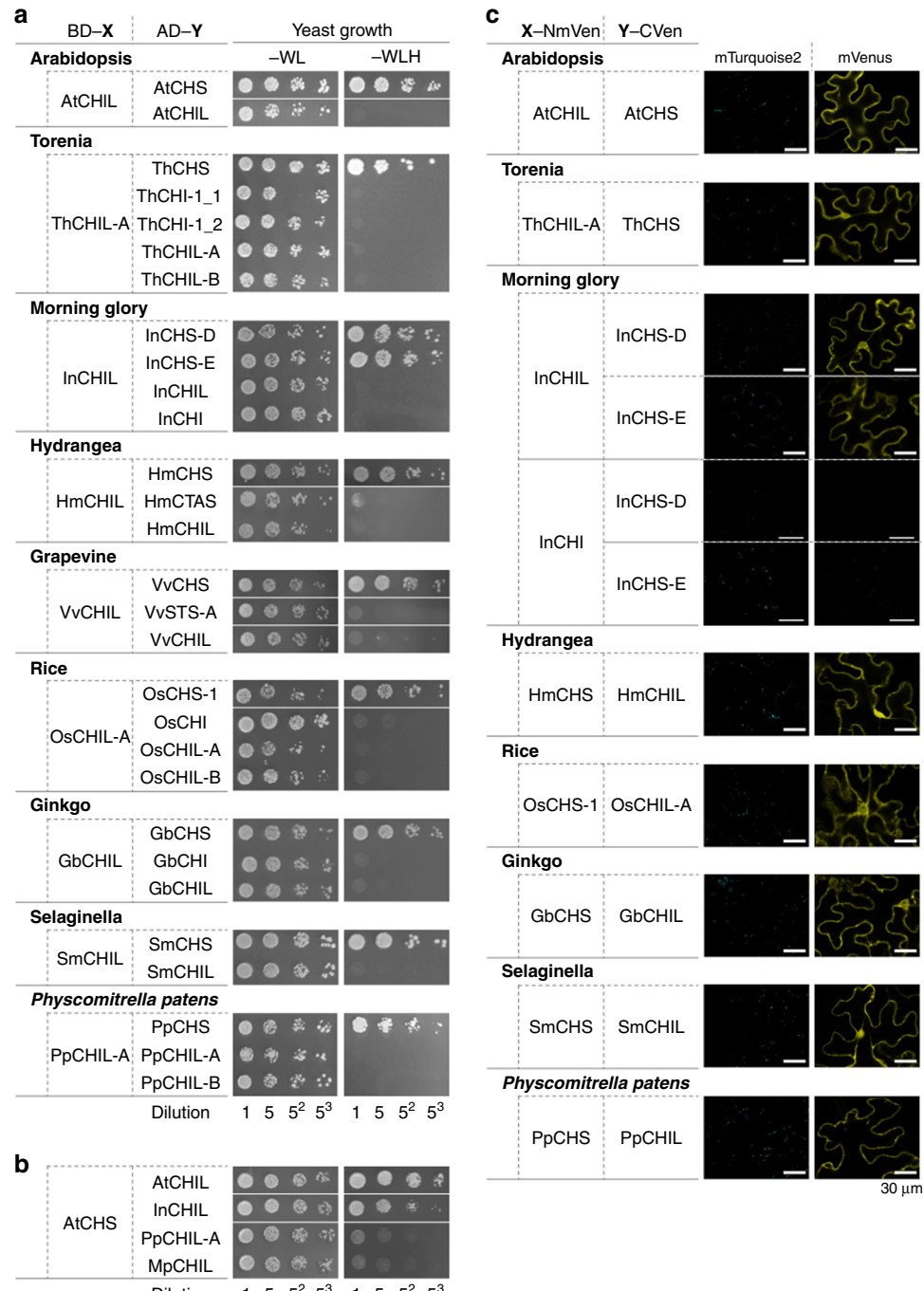

**Fig. 3 Binary interactions between CHIL and CHS in land plants. a** Interactions of CHIL with CHS and related proteins were analyzed in various land plants with the GAL4-based yeast two-hybrid system. HmCTAS and VvSTS refer to a *p*-coumaroyltriacetic acid synthase of hydrangea and a stilbene synthase isozyme of grapevine, respectively. Refer to the Fig. 2 legend for details regarding the other protein names and the abbreviated growth media names. **b** Interactions of heterologous CHILs with AtCHS. **c** Interactions of CHIL with CHS from various land plants were analyzed in *Nicotiana benthamiana* leaf cells by BiFC. The BiFC results for the CHI and CHS isozymes of morning glory (InCHI and InCHS-D/InCHS-E, respectively) are provided as negative controls. Scale bars = 30 μm.

diverse plant species[16,24], and the physical interactions between CHIL and CHS are strictly conserved among the examined land plants (Figs. 2–5, Supplementary Fig. 5, Supplementary Table 1). Moreover, the physical interactions between CHIL and CHS are highly specific. For example, the binding of CHIL to CHS takes place in a species-specific manner (Figs. 3b, 4). Moreover, although CHI is similar to CHIL regarding its primary and stereo structures[17], CHI does not necessarily bind to CHS (except in

snapdragon and *A. thaliana*; see refs. [7,9]) (Fig. 3a, c, Supplementary Fig. 5) or affect the product specificity of CHS (Fig. 5a, c). Additionally, although HmCTAS and VvSTS are very similar to CHS in terms of primary and stereo structures, the CHILs of *H. macrophylla* and *V. vinifera* cannot interact with these CHS-related enzymes (Fig. 3a, Supplementary Fig. 5) or affect their product specificities (Fig. 5c, d). Thus, on the basis of the observed ubiquitous occurrence and the conserved effect of CHIL on CHS in

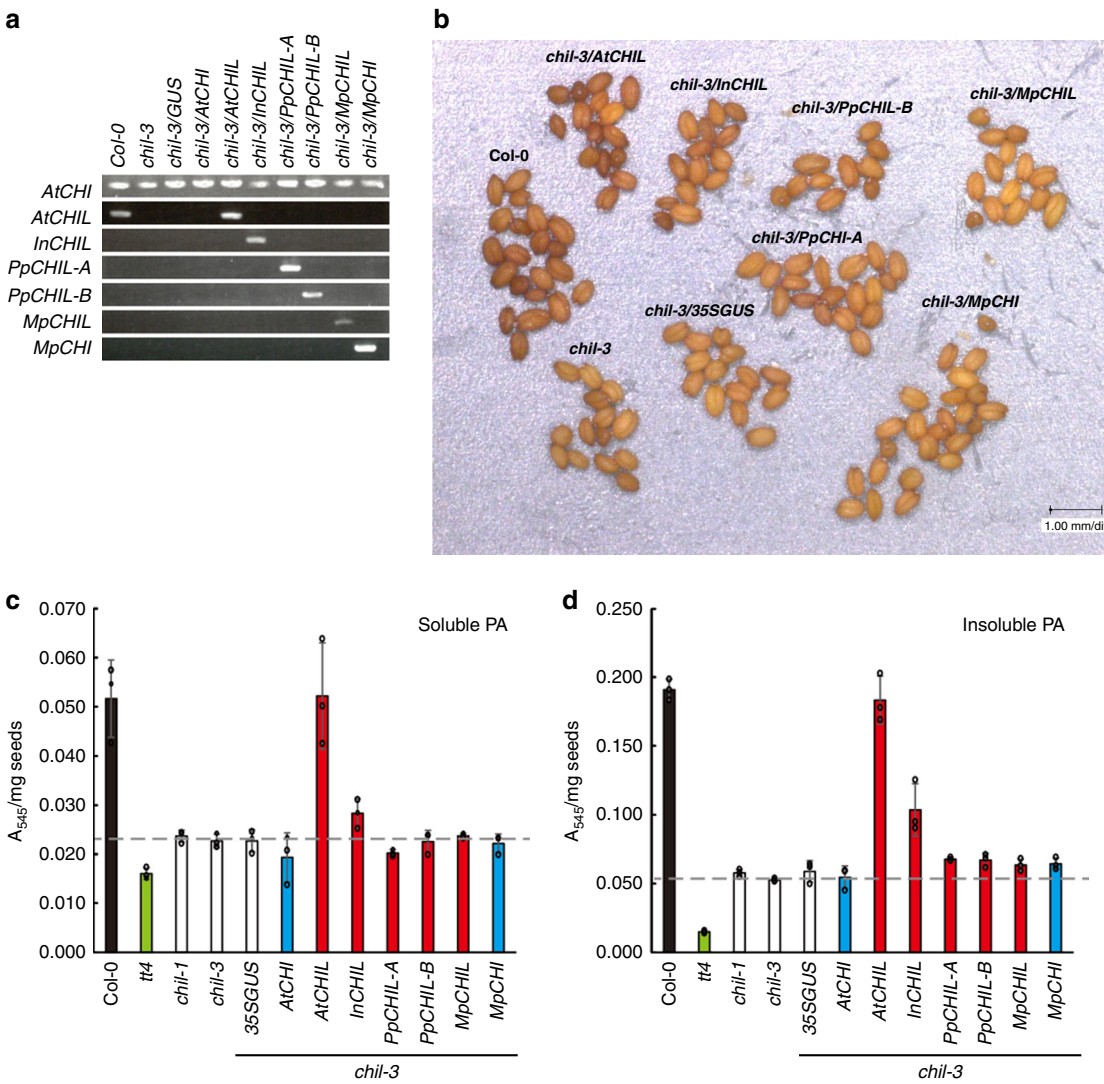

**Fig. 4 Complementation of in vivo CHIL function by heterologous CHILs in an *A. thaliana chil* mutant. a** RT-PCR analyses of the wild type (Col-0), the *chil-3* mutant, and the complemented *chil-3* mutants of *A. thaliana* for expression of the *CHIL* gene. **b** Seed phenotypes of Col-0, the *chil-3* mutant, and the complemented *chil-3* mutants. **c** Contents of soluble proanthocyanidins in mature seeds of Col-0 (black bar), the CHS mutant (*tt4*, green bar), the CHIL mutants (*chil-1* and *chil-3*; white bars), and the complemented *chil-3* mutants of *A. thaliana* with the *CHIL* gene (red bars), the *CHI* gene (light-blue bars) or the empty vector with *35SGUS* (white bar). **d** Contents of insoluble proanthocyanidins in mature seeds of the *A. thaliana* lines shown in (**c**). Data are presented as the average of three independent determinations ( ± standard deviation). The source data of the proanthocyanidin detrminations in (**c**) and (**d**) are provided in the Source Data file.

land plants, CHIL likely represents an auxiliary subunit of CHS that is essential for CHS to efficiently fulfill its role (chalcone synthesis) as the first committed enzyme of the flavonoid pathway. This, in turn, provides the possibility that among the CHS-related homologs, CHSs may be defined as members that interact with CHIL (see below).

The results of kinetic studies provided an important insight into the mechanistic aspects of the CHIL-mediated increase in the $E$ value of CHS-catalyzed reactions in flavonoid metabolons. The $k_{cat}$ values of THC production in the presence of CHIL were 2- to 15-fold greater than those in its absence in all examined plant species (Tables 2 and 3). The $K_m$ values for both substrates for THC production were also greater in the presence of CHIL than in its absence in all examined plant species (Tables 2 and 3). Thus, the $k_{cat}/K_m$ values of CHS enzymes were similar irrespective of the presence of CHIL. However, even though two enzyme systems have similar $k_{cat}/K_m$ values, the ratio of the rates of these

two systems may vary with the ratio of substrate concentration to $K_m$[25]. The $k_{cat}/K_m$ value only represents an estimation of the catalytic effectiveness of enzymes when substrate concentrations are near zero, which is unrealistic when the proposed involvement of CHS and CHIL in flavonoid metabolons is considered. In metabolons, a group of enzymes and metabolites in the metabolic pathway are believed to be concentrated in a small cellular region (i.e., micro-compartmentalization of cellular metabolism)[26]. Under such circumstances, the "effective" concentrations of *p*-coumaroyl-CoA and malonyl-CoA for CHS catalysis may be high enough relative to the $K_m$ values, and $k_{cat}$ may reflect the catalytic competency of the enzymes in metabolons relatively accurately. Therefore, the observed increase in the $E$ value of CHS-catalyzed reactions due to CHIL may be, at least partly, explained in terms of the CHIL-mediated enhancement of the $k_{cat}$ of THC synthesis because the $k_{cat}$ of THC synthesis may be related to the rate of step *d* (Fig. 1). CHIL increases the $E$ value of CHS catalysis via the

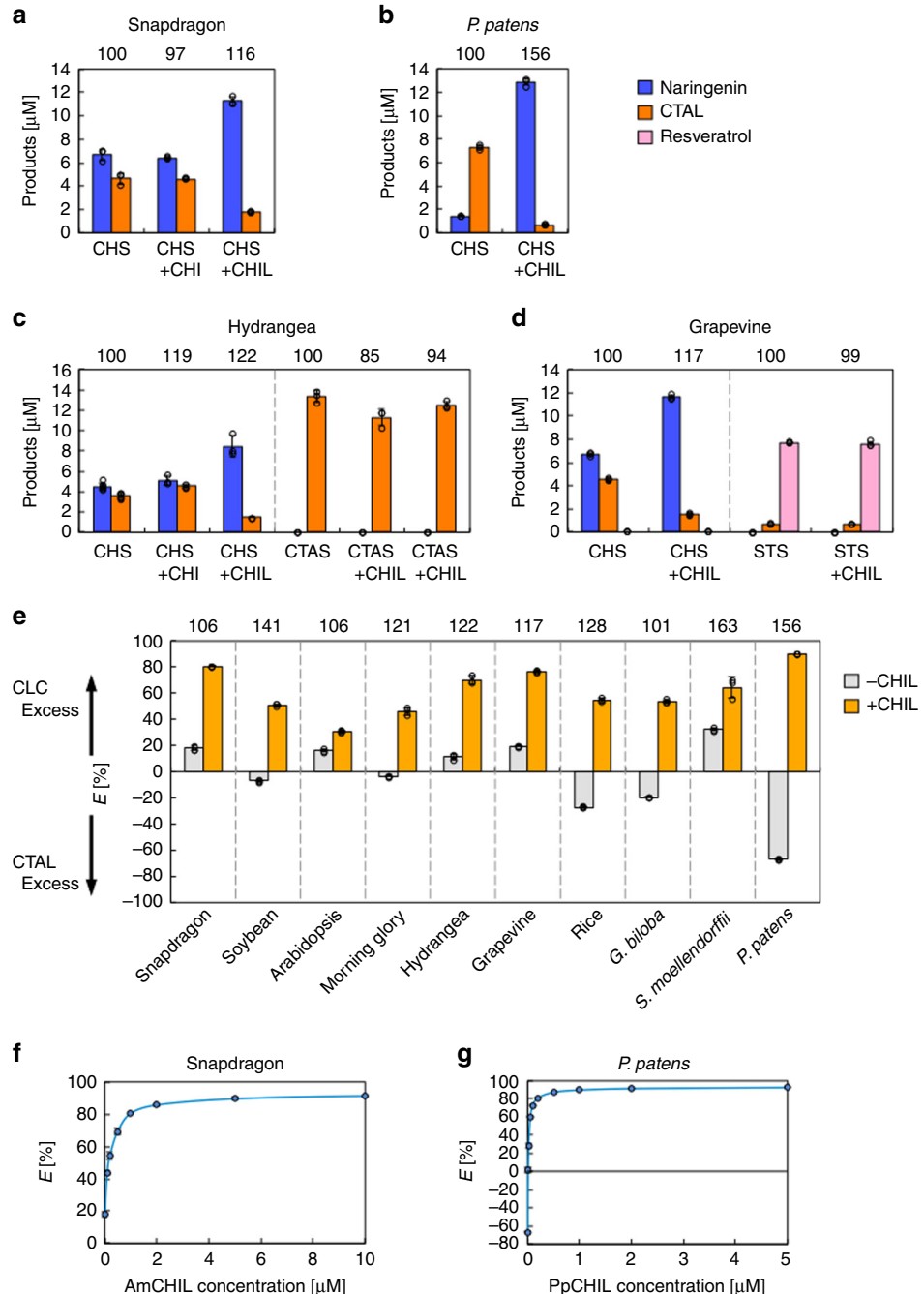

**Fig. 5 Effects of CHIL and related proteins on the product specificities of CHS, CTAS, and STS. a** Concentrations of products (blue bars, naringenin; orange bars, CTAL) formed after reactions catalyzed by 0.1 μM AmCHS in the presence and absence of 1 μM AmCHIL or AmCHI. **b** Concentrations of products (blue bars, naringenin; orange bars, CTAL) formed after reactions catalyzed by 0.1 μM PpCHS in the presence and absence of 1 μM PpCHIL. **c** Left, Concentrations of products formed after reactions catalyzed by 0.1 μM HmCHS in the presence and absence of 1 μM HmCHIL or HmCHI. Right, Concentrations of products formed after reactions catalyzed by 0.1 μM HmCTAS in the presence and absence of 1 μM HmCHIL or HmCHI. Blue bars, naringenin; orange bars, CTAL. **d** Left, Concentrations of products formed after reactions catalyzed by 0.1 μM VvCHS in the presence and absence of 1 μM VvCHIL. Right, Concentrations of products formed after reactions catalyzed by VvSTS in the presence and absence of VvCHIL. Blue bars, naringenin; orange bars, CTAL; and pink bars, resveratrol. **e** E values of reactions catalyzed by CHS (0.1 μM) in the presence (light-orange bars) and absence (gray bars) of homologous CHIL (1 μM) in various land plants. For (**a**–**e**), the numbers above panels indicate the relative percentages of the total amount of polyketide products, with the amount obtained in the absence of added CHIL or CHI set at 100%. **f** Effects of AmCHIL concentrations on the E values of the reaction catalyzed by 0.1 μM AmCHS. **g** Effects of PpCHIL concentrations on the E values of the reaction catalyzed by 0.1 μM PpCHS. For (**a**–**g**), enzymatic reactions were completed essentially under standard assay conditions (see Methods). Data are presented as the average of three independent determinations (±standard deviation). The underlying data for (**a**–**g**) are provided in the Source Data file.

**Table 2 Kinetic parameters for *p*-coumaroyl-CoA of THC production catalyzed by CHS of some land plants in the presence (+) and absence (−) of CHIL[a].**

| Enzyme | CHIL | Kinetics[b] | $k_{cat}$ (min⁻¹) | $K_m$ (μM) | $k_{cat}/K_m$ (min⁻¹ μM⁻¹) |
|---|---|---|---|---|---|
| PpCHS | + | c | [28.5 (±0.6)][c] | not determined[c] | 0.057[c] |
|  | − | H | 1.69 (±0.17) | 19.0 (±4.3) | 0.089 |
| SmCHS | + | H | 7.54 (±0.14) | 2.58 (±0.16) | 2.92 |
|  | − | H | 3.94 (±0.03) | 3.15 (±0.77) | 1.25 |
| GbCHS | + | H | 44.5 (±6.2) | 31.1 (±8.3) | 1.43 |
|  | − | H | 6.79 (±0.26) | 4.29 (±0.73) | 1.58 |
| OsCHS-1 | + | H | 19.4 (±3.3) | 76.3 (±19.0) | 0.25 |
|  | − | H | 1.58 (±0.06) | 7.70 (±0.93) | 0.21 |
| AmCHS | + | H | 76.3 (±6.0) | 49.7 (±6.4) | 1.54 |
|  | − | H | 5.23 (±0.17) | 4.73 (±0.58) | 1.11 |

[a]100 μM malonyl-CoA was used as the extender substrate.
[b]H, hyperbolic. Hyperbolic *v* vs [S] plots were obtained and fit to the Michaelis–Menten equation. For details, see Methods.
[c]A linear relationship between initial velocity and substrate concentrations [S] was obtained in the range of [S] examined (up to 50 μM), suggesting that the $K_m$ value of PpCHS for this substrate should be significantly greater than 50 μM, because the enzyme-catalyzed reactions proceeds with first-order kinetics under the conditions of [S] ≪ $K_m$. Therefore, only the $k_{cat}/K_m$ value was determined from slope of the initial velocity vs [S] plots. The $k_{cat}$ value was calculated using the initial velocity at [S] = 50 μM.

**Table 3 Kinetic parameters for malonyl-CoA of THC production catalyzed by CHS of some land plants in the presence (+) and absence (−) of CHIL[a].**

| Enzyme | CHIL | Kinetics[b] | $k_{cat}$ (min⁻¹) | K′ (μM) | $k_{cat}/K'$ (min⁻¹ μM⁻¹) | Hill coeff. |
|---|---|---|---|---|---|---|
| PpCHS | + | S | 31.5 (±0.6) | 9.14 (±0.36) | 3.45 | 2.96 (±0.32) |
|  | − | S | 2.60 (±0.12) | 3.23 (±0.22) | 0.80 | 3.25 (±0.85) |
| SmCHS | + | S | 23.5 (±0.4) | 3.86 (±0.14) | 6.09 | 2.11 (±0.14) |
|  | − | S | 5.62 (±0.21) | 1.15 (±0.09) | 4.90 | 2.71 (±0.52) |
| GbCHS | + | S | 28.9 (±1.7) | 8.97 (±0.80) | 3.22 | 2.46 (±0.45) |
|  | − | S | 8.01 (±0.36) | 3.60 (±0.22) | 2.22 | 3.03 (±0.59) |
| OsCHS-1 | + | S | 7.03 (±0.36) | 7.60 (±0.58) | 0.93 | 2.31 (±0.34) |
|  | − | S | 2.41 (±0.11) | 3.35 (±0.16) | 0.72 | 3.00 (±0.51) |
| AmCHS | + | S | 35.4 (±1.4) | 12.2 (±1.6) | 2.90 | 1.73 (±0.31) |
|  | − | S | 8.47 (±0.14) | 2.57 (±0.10) | 3.30 | 3.77 (±0.73) |

[a]50 μM *p*-coumaroyl-CoA was used as the starter substrate.
[b]S, sigmoidal. Sigmoidal *v* vs [S] plots were obtained and fit to the Hill's equation. For details, see Methods.

specific rate enhancement of step ***d***, resulting in the increased channeling of **3** toward THC production at the expense of CTAL formation (step ***e***) (Fig. 1).

We revealed that CHIL is a non-catalytic component of flavonoid metabolons (Supplementary Fig. 1), suggesting that the importance of metabolon formation in plant specialized metabolism includes its role to rectify promiscuity of metabolic enzymes via macromolecular interactions to decrease undesirable minor activities and/or specificities. This potential consequence of macromolecular interactions in metabolons is based on the finding that the promiscuous product specificity of *Rhododendron dauricum* orcinol synthase, which is a type III PKS involved in orsellinic acid biosynthesis, may be rectified by the presence of an olivetolic acid cyclase from another plant species (*Cannabis sativa*)[27]. Thus, the rectification of catalytic promiscuity via macromolecular interactions might not be limited to the CHIL–CHS system and may be generally important for metabolons involved in plant specialized metabolism. This strategy may enable the maintenance of the native promiscuity of enzyme catalysis for the evolution of functions, and facilitate the efficient synthesis of specialized metabolites based on the rectified specificity in extant plants.

The suppression of CHIL reportedly results in a 3- to 9-fold decrease in flavonoid contents in petunia and torenia petals[16]. The in vitro effects of CHIL on the product specificity of CHS

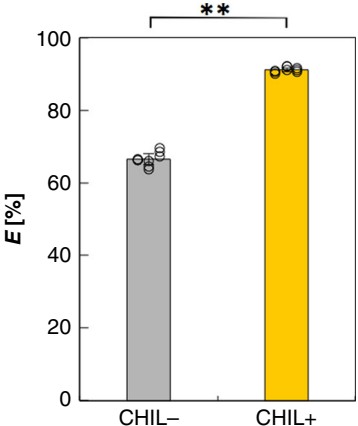

**Fig. 6 Effects of CHIL on in vivo specificity of CHS.** *E* values of CHS-catalyzed reactions during the production of CLC from *p*-coumaric acid in *E. coli* cells harboring pET15b-Gm4CL-3-GmCHS-1 (CHIL−) or pET15b-Gm4CL-3-GmCHS-1-GmCHIL (CHIL+) are shown. Data are presented as the average of three independent determinations of three biological replicates (±standard error). \*\**P* < 0.01 (Student's *t*-test). The underlying data are provided in the Source Data file.

clarified in this study may, at least partly, account for such an in vivo effect of CHIL. However, CHIL appears to have another important role in enhancing flavonoid production in a plant species-specific manner. Specifically, the results of the present protein–protein interaction analyses proved that in snapdragon and soybean, CHIL physically interacts with CHS as well as cytochromes P450 (i.e., FNSII in snapdragon and IFS in soybean; see Supplementary Fig. 1), which also bind to CHI in these plant species[8–10]. Additionally, in *A. thaliana*, which lacks *FNSII* and *IFS* in its genome, CHIL was observed to interact with CHI[18]. It is generally accepted that cytochromes P450 serve as the nuclei of metabolon formation in plant specialized metabolism[28]. Thus, these observations imply there is another important CHIL function related to metabolon formation. Specifically, it recruits enzymes involved in the early flavonoid biosynthesis stage (CHS and CHI) to a flavonoid biosynthesis site in plant cells to establish an efficient supply of naringenin, which is a common precursor for the biosynthesis of flavonoids further down the pathway (Fig. 1). Therefore, both of these roles (i.e., rectification of catalytic promiscuity and recruiting enzymes to a flavonoid biosynthesis site) might be important for CHIL to fully enhance production in planta, with the latter occurring in a species-specific manner.

An evolutionary connection between fatty acid metabolism and flavonoid biosynthesis has been proposed based on the structural and mechanistic similarities between the enzymes involved in the biosynthesis of these compounds[17,29]. Consistent with this, CHI and CHILs are phylogenetically related to FABP. More specifically, CHILs are members of the CHI-fold protein family, which includes the following four phylogenetically separate groups of proteins: CHI enzymes (types I and II), FABPs (type III), and CHILs (type IV). In planta, FABP localizes to plastids, where it is involved in de novo fatty acid biosynthesis. A phylogenetic analysis suggests that FABPs may be the oldest members of the CHI-fold protein family and CHIL and CHI may have evolved from FABP when plants colonized the terrestrial environment[17,29]. In this context, it would be noteworthy that the diffusible linear tetraketide-CoA intermediate [**3** (R = CoA), see Fig. 1] formed during CHS catalysis is structurally and biosynthetically related to fatty acids, and computational docking studies predicted that CHIL potentially binds to the intermediate **3** in an energetically favorable manner (Supplementary Fig. 9). This is consistent with the known abilities of other types of proteins in the family to bind to fatty acids (for FABPs) and cyclic tetraketides (for CHIs)[17,29]. This prediction based on an evolutionary consideration should be addressed in conjunction with the mechanistic aspects of the role of CHIL to rectify CHS promiscuity in future studies.

Phylogenetic analysis suggested that CHS also evolved when plants colonized the terrestrial environment, similar to CHIL. Unlike the CHI-fold protein family, however, CHS constitutes a multigene family that consists of species-specific copies of CHS-related genes (Supplementary Fig. 10). Notably, four clades were resolved that each consist of multiple CHS genes from a single species (*M. polymorpha, S. fallax, P. patens,* and soybean) (Supplementary Fig. 10a, c). This finding suggests that recent gene duplications have occurred within these species or a recent ancestor. It is also likely that these CHS-related genes evolved via a birth-and-death process, in which new copies of CHS genes are produced by repeated gene duplication and some duplicate genes are maintained in the genome while others are deleted, neofunctionalized by mutations, or rendered nonfunctional by deleterious mutations. We included the STS and CTAS sequences in the phylogenetic analysis to infer their evolutionary origin. The analysis indicated that STS (PsSTS and VvSTS) and CTAS (HmCTAS) were differentiated from CHS genes of seed plants

(gymnosperms and angiosperms) in a lineage-specific manner (Supplementary Fig. 10b, d), thus functional differentiation of STS and CTAS from CHS arose during seed plant evolution. Our functional analysis of CHIL predicted that ancestral CHS had the ability to interact with CHIL to facilitate its biochemical role of THC synthesis. STS and CTAS did not interact with CHIL. Thus, during the proposed evolutionary process of CHS-related genes, some CHS homologs may have lost their ability to interact with CHIL, allowing neofunctionalization to produce enzymes that are capable of exclusively producing *p*-coumaroyltriacetic acid (e.g., HmCTAS) and stilbenes (e.g., VvSTS). This is consistent with the notion that the ability of CHS to produce THC was maintained with continued ability to interact with CHIL throughout the course of land plant evolution.

In conclusion, CHIL is a rectifier of promiscuous CHS that allows CHS to precisely fulfill its physiological function—the synthesis of THC, which is the key precursor of a diverse array of flavonoids indispensable for the adaptation of land plants to the terrestrial environment. Given that this role of CHIL has been maintained during the evolution of land plants, it must have been of vital importance to survival of land plants in the terrestrial environment.

## Methods

**Plasmids, *E. coli* strains, and *A. thaliana* lines**. The plasmids used in this study were obtained from the following sources: pGEM-T Easy vector (Promega, Madison, WI, USA); pGBKT7 and pGADT7 (each used for the GAL4-based yeast two-hybrid assays) and pET15b and pCold I (each used for the heterologous production of His6-tagged proteins in *E. coli* cells) (Takara Bio, Otsu, Japan); pCOLADuet-1 (used for the simultaneous heterologous production of His6-tagged proteins and S-tagged proteins in *E. coli* cells) (Merck, Darmstadt, Germany); pGEX4T-1 (used for the heterologous production of glutathione S-transferase-tagged proteins in *E. coli* cells) (GE Healthcare Japan, Tokyo, Japan); pDOE-05[30], ER-rb[31] (each used for BiFC experiments), and pGWB1 (The Arabidopsis Biological Resource Center, Columbus, OH, USA). *Escherichia coli* strains BL21 and BL21 (DE3) were obtained from Promega and strain Rosetta2 (DE3) was obtained from Merck KGaA. *Arabidopsis thaliana* accession Columbia-0 (Col-0; Lehle Seeds, http://www.arabidopsis.com) was used as the wild type in this study. The T-DNA insertion mutants of *Arabidopsis*, lines SALK_096551 (*chil-1*)[18] and GABI_189B03 (CS745226, *chil-3*) for CHIL were obtained from the Arabidopsis Biological Resource Center. T-DNA insertion lines were screened by PCR using specific primers (CHIL-F and CHIL-R for *CHIL* and *TF* and *TR* for T-DNA; see Supplementary Table 2). PCR products were sequenced to determine the exact insertion points.

**Chemicals**. The THC and *p*-coumaroyl-CoA were purchased from TransMIT (Gießen, Germany). Malonyl-CoA, naringenin, *trans-p*-coumaric acid, resveratrol, ampicillin, imidazole, and isopropyl 1-β-D-thiogalactoside were purchased from Nacalai Tesque (Kyoto, Japan). Acetonitrile and trifluoroacetic acid were obtained from Sigma-Aldrich Japan (Tokyo, Japan). Bis(sulfosuccinimidyl) suberate disodium salt was obtained from Dojindo Laboratories (Kumamoto, Japan), whereas $Ni^{2+}$-coated magnetic beads (His Mag Sepharose Ni) and $Ni^{2+}$-coated Sepharose beads (His SpinTrap) were purchased from GE Healthcare Japan. All chemicals were of analytical grade except for acetonitrile and trifluoroacetic acid, which were of HPLC grade. [2-14C]Malonyl-CoA (1480 MBq/mmol) was purchased from PerkinElmer Life Sciences (Boston, MA, USA).

**Synthesis of cDNAs encoding flavonoid enzymes and proteins**. The cDNAs encoding flavonoid enzymes and CHILs analyzed in this study are summarized in Supplementary Table 4. The cDNAs encoding flavonoid enzymes of snapdragon, torenia, and soybean [i.e., CHS, chalcone reductase (CHR, soybean), CHI, 2-hydroxyisoflavanone dehydratase (HID, soybean), FNSII (snapdragon and torenia), IFS (soybean), flavanone 3-hydroxylase (F3H, snapdragon), and dihydroflavonol 4-reductase (DFR, snapdragon)] were obtained as previously described[8–10]. The CHIL cDNAs of snapdragon[32], torenia[16], soybean[8,10], *A. thaliana*[18], rice (*Oryza sativa*, OsCHIL-A and OsCHIL-B)[19], and morning glory[16] were obtained as described in the listed references.

The *CHIL* genes of grapevine (*VvCHIL*), *Selaginella moellendorffii* (*SmCHIL*), and *P. patens* (*PpCHIL-A* and *PpCHIL-B*) were identified via a BLAST search of the Phytosome (version 12.1) database (https://phytozome.jgi.doe.gov/pz/portal.html), with the *AtCHIL* sequence as a query (see Supplementary Table 4). The *CHIL* and *CHI* genes of hydrangea (*HmCHIL* and *HmCHI*) were identified via a BLAST search of the Hardwood Genomics Project database

(https://www.hardwoodgenomics.org), with the *AtCHIL* and *AtCHI* sequences as queries, respectively. The *S. moellendorffii CHS* gene (*SmCHS*) was identified via a BLAST search of the Phytosome (version 12.1) database, with the *AtCHS* sequence as a query. The gingko *CHIL* gene (*GbCHIL*) was identified via a local blast search of a CDS fasta file from GigaDB (https://www.gigadb.org/dataset/100209). The *CHIL* and *CHI* genes of *M. polymorpha* (*MpCHIL* and *MpCHI*) were identified via a BLAST search of the MarpolBase database, with the *AtCHIL* and *AtCHI* sequences as queries, respectively.

The cDNAs encoding the CHS, CHI, and related enzymes of the following plant species were obtained essentially as previously described: rice (*OsCHS-1*, *OsCHS-2*, *OsCHI*)[33], gingko (*GbCHS* and *GbCHI*)[34], *P. patens* (*PpCHS*)[35], and hydrangea (*HmCHS* and *HmCTAS*)[15].

**Yeast two-hybrid assays with the split-ubiquitin system**. The interactions between CHILs and cytochromes P450 from snapdragon and soybean (AmFNSII, GmIFS-1, and GmIFS-2) were analyzed in the split-ubiquitin system with the DUALmembrane Kit 3 (Dualsystems Biotech, Zurich, Switzerland) essentially as previously described[8–10]. Briefly, *Saccharomyces cerevisiae* strain NMY51, which was included in the kit, was transformed with one of the following pairs of plasmids (i.e., derivatives of pBT3-SUC and pPR3-N; derivatives of pBT3-SUC and pPR3-C; or, as a positive control, pOst1-NubI and derivatives of pBT3-SUC). The transformant cells were grown on agar plates of SD/–WL (SD lacking tryptophan and leucine), SD/–WLH (SD lacking tryptophan, leucine, and histidine), and SD/–WLHAde (SD lacking tryptophan, leucine, histidine, and adenine) at 30 °C for 2–4 days.

**GAL4-based yeast two-hybrid assays**. The Matchmaker Gold Yeast Two-Hybrid System (Clontech, Mountain View, CA, USA) was used to complete GAL4-based yeast two-hybrid assays as previously described[8–10]. Briefly, the open reading frames encoding proteins to be tested for interaction were amplified by PCR to add SfiI restriction sites for ligation with the pGBKT7 and pGADT7 vectors (Clontech). Each amplified DNA was purified, digested with SfiI, and inserted into the SfiI site of the pGBKT7 or pGADT7 vectors. The *S. cerevisiae* strain Y2H Gold (Clontech) was transformed with each pair of the plasmids in accordance with the manufacturer's guidelines.

**BiFC**. For BiFC assays, the binary vector pDOE-05[30] was used to express the proteins of interest, which were fused with NmVen210 and CVen210 (see Results). For example, *AmCHS* cDNAs were digested with *NcoI/SpeI* and ligated into multiple cloning site (MCS) 1 of pDOE-05 to generate pDOE05-AmCHS. The *PpuMI/AatII*-digested *AmCHIL* cDNA was then inserted into the *SanDI/AatII* sites in MCS3 of these plasmids to produce pDOE05-AmCHS-AmCHIL. The binary plasmid ER-rb[31] was used to express an ER marker protein (mCherry-HDEL). *Agrobacterium tumefaciens* GV3101(pMP90) cells harboring one of the pDOE derivatives were used to transform the leaves of wild-type *Nicotiana benthamiana* plants via a previously described agroinfiltration procedure[8,9]. The plants were then incubated at 25 °C under long-day conditions for 2 days. Fluorescence in tobacco leaf cells was observed with a TCS-SP8 confocal laser scanning microscope (Leica, Mannheim, Germany) comprising a white light laser and a HyD detector as previously described[8,9].

**Co-precipitation experiments**. (Method I) The crude extract of the orange snapdragon petals (cv. Maryland Orange) were mixed with Ni$^{2+}$-coated magnetic beads, after which one of the bait proteins (His$_6$-AmCHIL and His$_6$-AmCHS) was added and the mixture was incubated at room temperature for 1 h. The beads were recovered and washed with 0.01 M potassium phosphate buffer, pH 7.0, containing 0.15 M NaCl. The bead-bound proteins were then eluted by washing the beads with 0.1 M glycine-HCl buffer at pH 2.8. The eluted proteins were digested with trypsin and the resulting peptides were recovered by solid-phase extraction followed by sequencing by LC-MS/MS. The peptides were identified by mapping the derived peptide sequences against the snapdragon genome sequence[36].

(Method II) His$_6$-AmCHIL was added to the extract (300 μl) from the red petals (stages 1 through 6) of snapdragon cv. Montego Red (Dainichi Shokai; Koga, Ibaraki, Japan). One-hundred microliters of Ni$^{2+}$-coated Sepharose beads (His SpinTrap) were added to the mixture, which was then incubated at 4 °C for 1 h, followed by centrifugation at 100 × *g* for 1 min. The supernatant (termed fraction F) was recovered. The beads were washed three times with 300 μl of 0.05 M HEPES-NaOH, pH 7.5, and the third-wash supernatant (termed fraction W) was recovered. The bead-bound proteins were then eluted by washing the beads with 0.05 M HEPES-NaOH, pH 7.5 containing 500 mM imidazole (termed fraction E). SDS-PAGE and western blotting analyses were carried out as described previously[9] using anti-AmCHS immunoglobulin G (IgG) as a primary antibody. For the control, His$_6$-AmCHIL was replaced by water.

**Complementation tests using an *A. thaliana* mutant**. For complementation tests, 1.7-kb genomic fragments covering 1696 bp of the *AtCHIL* promoter region were amplified by PCR using the primers AF and AR (see Supplementary Table 2). The amplified fragment was cloned into the pENTR/D-TOPO vector (ThermoFisher Scientific, Waltham, MA, USA) to construct the plasmid pKYS453 and sequenced

to confirm the absence of PCR errors. The full-length coding regions of *AtCHI* and *AtCHIL* were amplified by PCR using specific primers (BF and BR for *AtCHI* and CF and CR for *AtCHIL*) and fused to pKYS453 using the In-Fusion HD Cloning Kit (Takara Bio, Shiga, Japan) to yield the entry vectors pKYS454 and pKYS455, respectively (Supplementary Tables 2 and 3). The full-length coding regions of *PpCHIL-A*, *PpCHIL-B*, *InCHIL*, *MpCHIL*, and *MpCHI* were amplified by PCR using specific primers (DF and DR for *PpCHIL-A*, EF and ER for *PpCHIL-B*, FF and FR for *InCHIL*, GF and GR for *MpCHIL*, and HF and HR for *MpCHI*) and fused to pKYS453 using the In-Fusion HD Cloning Kit to yield the entry vectors pKYS459, pKYS458, pKYS462, pKYS463, and pKYS464, respectively (Supplementary Tables 2 and 3).

pGWB1 was used as a destination vector, and the LR reactions with the entry vectors prepared as above were catalyzed by the Gateway LR Clonase Enzyme mix (ThermoFisher Scientific) to obtain the binary vectors pKYS456 for *AtCHI*, pKYS457 for *AtCHI*, pKYS461 for *PpCHIL-A*, pKYS460 for *PpCHIL-B*, pKYS465 for *InCHIL*, pKYS466 for *MpCHIL*, and pKYS467 for *MpCHI*. Transformation of *Agrobacterium tumefaciens* and *Arabidopsis thaliana* (*chil-3*) and the selection of transformants were carried out as described previously[37].

**RT-PCR**. Total RNA was extracted using the RNAqueous Total RNA Isolation Kit (ThermoFisher Scientific) in accordance with the manufacturer's instructions. After DNase I treatment using the TURBO DNA-*free* Kit (ThermoFisher Scientific), 500 ng total RNA was used for cDNA synthesis using the SuperScript™ III First-Strand Synthesis System (Invitrogen). RT-PCR was performed using TaKaRa Ex Taq™ (Takara Bio) with a denaturation step at 98 °C for 2 min, followed by 35 cycles of amplification (98 °C for 10 s, 55 °C for 30 s, and 72 °C for 2 min), and a final extension step (72 °C for 5 min). The primers AtCHI_RT_F and AtCHI_RT_R for *AtCHI*, At5g05270_359F and AtCHIL_RT_R for *AtCHIL*, Ipom_EFP_RT_F and Ipom_EFP_RT_R for *InCHIL*, EFPa105_RT_F and EFPa105_RT_R for *PpCHIL-A*, EFPb104_RT_F and EFPb104_RT_R for *PpCHIL-B*, poly_EFP_RT_F and poly_EFP_RT_R for *MpCHIL*, and poly_CHI_RT_F and poly_CHI_RT_R for *MpCHI* were used for analyses (Supplementary Table 2).

**PA analysis**. Extraction and acid hydrolysis of PA were performed in triplicate as described previously[38,39]. Mature seeds (10 mg) were homogenized in 0.75 ml of 70% (v/v) acetone containing 5.26 mM Na$_2$S$_2$O$_5$ in a mixer mill (Qiagen Retsch MM300 TissueLyser; Qiagen, Tokyo, Japan) at 20 Hz for 1 min, followed by sonication for 20 min. After centrifugation at 15,000 × *g* for 5 min, the supernatant was evaporated and resuspended in 1 ml of a 2:10:3 (v/v/v) mixture of HCl:butanol:70% acetone. The absorbance of the solutions before and after hydrolysis at 95 °C for 60 min was measured at 545 nm and the difference was treated as the soluble PA fraction. The pellet after extraction with 70% acetone was also evaporated, suspended in the mixture of HCl:butanol:70% acetone, and hydrolyzed, and was treated as the insoluble PA fraction.

**Tandem mass spectrometry analyses**. The amino acid sequences of the peptides obtained during the co-precipitation experiments (Supplementary Fig. 3) were determined with an EASY-n LC1000 system equipped with the Q Exactive Plus apparatus (ThermoFisher Scientific). The products of the AmCHS-catalyzed reaction (Supplementary Fig. 7) were identified with the Q Exactive Plus apparatus.

**Expression and purification of CHS, CHIL, and other enzymes**. The complete coding sequences for *CHS*- and *CHIL*-related genes (i.e., *CHS*, *CTAS*, *STS*, *CHI*, and *CHIL*) were amplified by PCR, with the pGEM-T Easy-, pGKT7-, or pBluescript-based constructs as templates. Restriction enzyme sites were added during the PCR amplification for a subsequent subcloning into the pCold I vector. The resulting fragments were subcloned into the pCold I vector to express a fusion protein with an *N*-terminal His$_6$ tag. *Escherichia coli* BL21 cells were transformed with the resulting plasmids. The heterologous expressions of these cDNAs and the purification of the recombinant proteins were completed as previously described[8–10]. The purified proteins were analyzed by SDS-PAGE and visualized with Coomassie Brilliant Blue R250. The concentrations of the recombinant proteins were determined based on the absorption coefficients at 280 nm, which were calculated based on the amino acid sequences[40].

**Enzyme assays**. The standard assay mixture consisted of 100 mM HEPES-NaOH buffer, pH 7.5, 50 μM *p*-coumaroyl-CoA, 100 μM malonyl-CoA, CHS (typically 0.1 μM), and CHIL (typically 1.0 μM) in a final volume of 50 μl. The mixture without CHS was pre-incubated at 30 °C for 5 min, and the reaction was started by adding CHS. After a 60-min incubation at 30 °C, the reaction was stopped by adding a 50-μl aliquot of a mixture comprising acetonitrile and water (2:3, v/v) and 4% (v/v) trifluoroacetic acid. Flavonoids and related products formed in the reaction mixture were analyzed by reversed-phase HPLC with a J'Sphere ODS M80 column (4.6 × 150 mm; YMC, Kyoto, Japan) under previously described conditions[10]. Chromatograms were obtained by monitoring the absorbance at 290 nm. In some cases, flavonoids and related products were also analyzed by LC-MS with the following Shimadzu LCsolution system: column, CAPCELL CORE C18 (2.1 × 100 mm; Shiseido, Tokyo, Japan); flow rate, 0.2 ml/min; solvent A, 0.05% (by volume)

formic acid in $H_2O$; solvent B, 0.05% (by volume) formic acid in acetonitrile. After an injection (5 μl) onto the column that was equilibrated with 20%B (by volume), the column was initially developed isocratically with 20%B for 2 min, followed by a linear gradient from 20 to 35%B in 18 min. The column was then washed isocratically with 35%B for 5 min, followed by a linear gradient from 35 to 90%B in 1 min. The column was washed isocratically with 20%B for 10 min before the next injection to ensure the column was sufficiently re-equilibrated. Chromatograms were obtained via mass spectrometry (SIM, negative ion mode).

To quantify the CTAL formed, the standard assay mixture was supplemented with [2-$^{14}$C]malonyl-CoA (148 MBq/mmol). The CTAL and naringenin were separated by HPLC as described above[10] and collected, after which the radioactivities of the CTAL and naringenin fractions were determined with the LS6500 Multi-Purpose Scintillation Counter (Beckman Coulter, Brea, CA, USA). The radioactivity of the naringenin fraction was correlated with absorbance peak integrals of known amounts of naringenin, whose specific radioactivity was equal to that of CTAL.

**Enzyme kinetics.** Assays of the initial velocity ($v$) of the CHS-catalyzed reactions were completed with a standard assay system (see above) under steady-state conditions and with various substrate concentrations. The apparent $K_m$ and $V_{max}$ values and their standard errors for $p$-coumaroyl-CoA in the presence of a fixed concentration of malonyl-CoA were determined by fitting the initial velocity data to the Michaelis–Menten equation:

$$v = \frac{V_{max}[S]}{K_m + [S]}$$

with a nonlinear regression method of the SigmaPlot 12 program (Hulinks, Tokyo, Japan), where $V_{max}$, $K_m$, and [S] denote maximum velocity, the Michaelis constant, and the substrate concentration, respectively.

When CHS assays were completed with malonyl-CoA as the varying substrate with a fixed concentration of $p$-coumaroyl-CoA, sigmoidal $v$ vs [S] plots were obtained and fit to Hill's equation:

$$v = \frac{V_{max}[S]^n}{K' + [S]^n}$$

with a nonlinear regression method of the SigmaPlot 12 program, where $K'$ is related to $K_m$ and represents the effect of the substrate occupancy at one site on the substrate affinity at other sites, and $n$ corresponds to Hill's coefficient.

**Biolayer interferometry.** Biolayer interferometry measurements were recorded with a BLItz instrument (ForteBio, Fremont, CA, USA). Biosensors were soaked in BLItz assay buffer (100 mM HEPES-NaOH, pH 7.5) for 10 min, and His$_6$-tagged CHS was immobilized on anti-His$_6$ sensors. Biolayer interferometry assays consisted of the following three steps, all performed in BLItz assay buffer: step 1, initial baseline (0–30 s); step 2, loading of glutathione S-transferase-tagged CHIL (30–150 s); and step 3, wash (150–270 s). Control values, which were obtained from empty sensors (i.e., no protein), were subtracted from experimental values before the data were processed. To calculate the equilibrium dissociation constant ($K_D$) and the association ($k_a$) and dissociation ($k_d$) rate constants, sensorgrams were fit to a 1:1 binding model with BLItz Pro (version 1.2.1.2).

**Production of chalcone in *E. coli* cells.** The *Gm4CL-3* cDNA (DDBJ/EMBL/GenBank accession number, AF002258) was amplified by PCR from a cDNA library prepared from the lateral roots of soybean seedlings (cv. Enrei) as described previously[8] and cloned into *Nde*I/*Xho*I-digested pET-15b according to the SLiCE method[41] to obtain pET-15b-Gm4CL-3. The *GmCHS-1* cDNA was amplified by PCR and cloned into MCS1 of *Eco*R1-digested pCOLADuet-1 to obtain pCOLADuet-1-GmCHS-1. The DNA segment for the T7 promoter-driven expression of *GmCHS-1* in the resulting plasmid was then introduced into *Eag*I-digested pET15b-Gm4CL-3 to obtain pET15b-Gm4CL-3-GmCHS-1. The *GmCHIL* cDNA was amplified by PCR and cloned into MCS2 of *Nde*I-digested pCOLADuet-1-GmCHS-1 to obtain pCOLADuet-1-GmCHS-1-GmCHIL. The DNA segment for the T7 promoter-driven co-expression of *GmCHS-1* and *GmCHIL* in the resulting plasmid was then introduced into *Eag*I-digested pET15b-Gm4CL-3 to obtain pET15b-Gm4CL-3-GmCHS-1-GmCHIL. The pET15b-Gm4CL-3-GmCHS-1 and pET15b-Gm4CL-3-GmCHS-1-GmCHIL recombinant plasmids were respectively used to transform *E. coli* strain Rosetta2 (DE3) to obtain the strains CHIL– and CHIL+, respectively. The transformed cells were grown on Luria-Bertani (LB) medium containing 50 μg/ml kanamycin and 34 μg/ml chloramphenicol.

The transformed cells were grown in 20 ml LB medium containing 50 μg/ml ampicillin and 50 μg/ml chloramphenicol at 37 °C with shaking until the optical density at 600 nm (OD$_{600}$) reached 0.4–0.5. Isopropyl 1-β-D-thiogalactoside was then added to the culture at a final concentration of 1.0 mM, after which the cells were incubated at 18 °C for 20 h. Cells were harvested by centrifugation, washed twice with M9 medium, and suspended in M9 medium for an OD$_{600}$ of 0.5 (final volume, 200 μl). *p*-Coumaric acid was added to the cell suspension for a final concentration of 1 mM and the mixture was incubated at 30 °C for 3 h and then

centrifuged. The target products (CLC, CTAL, and BNY) in the supernatant and precipitate fractions were extracted with a 2:1 (v/v) mixture of methanol:chloroform and analyzed by LC/MS as described above.

To quantify the expressed GmCHS-1 and GmCHIL proteins, in which GmCHS-1 was His$_6$-tagged while GmCHIL was not, the crude extracts of the engineered *E. coli* cells, as well as known amounts of GmCHS-1 and GmCHIL proteins, were subjected to SDS-PAGE followed by western blotting analyses using anti-His$_6$ IgG (to determine GmCHS-1) and anti-CHIL IgG (to determine GmCHIL). The immunoblots were quantified using ImageJ software (National Institutes of Health, Bethesda, MD, USA).

**Preparation of antibodies.** Anti-AmCHS antibodies were prepared as described previously[9]. To obtain anti-CHIL antibodies, the peptides corresponding to amino acid residues 111–125 of AmCHIL, which were predicted to serve as epitopes, were used to immunize female Japanese white rabbits. The initial injection was administered in complete Freund's Adjuvant with all subsequent immunizations in incomplete Freund's Adjuvant. Immunoglobulin G (IgG) was purified from the collected serum as described previously[9]. Monoclonal anti-His$_6$ antibody was obtained from Wako, Osaka, Japan (Code: 011-23091, Lot. LKQ2757).

**Phylogenetic analysis.** The amino acid sequences of CHS and their homologs were used as queries to search a nucleotide sequence database of protein-coding genes from nine plant species (*A. thaliana*, *G. max*, *V. vinifera*, *O. sativa*, *S. moellendorffii*, *S. fallax*, *M. polymorpha*, *P. patens*, and *C. reinhardtii*) using the tblast program. Sequences were retrieved from the Phytoszome (version 12.1) database. *Pinus sylvestris* STS and CHS amino acid sequences were obtained from the original publications[42,43]. Matched sequences with e-value > 0.01 or match length <350 bp were eliminated. After merging the query sequences with the matched sequences, a multiple alignment of the amino acid sequences was generated using MUSCLE with the default settings. Phylogenetic analyses were performed using the neighbor-joining (NJ) method[44] and maximum likelihood (ML) method[45] as implemented in SEA-VIEW version 4.7 and RAxML-NG version 0.9[46], respectively. Poisson corrected evolutionary distance was used for NJ tree reconstruction. An LG amino acid substitution matrix[47] with gamma model rate heterogeneity and empirical amino acid frequencies was used for the ML analysis. Confidence of inferred phylogenetic trees were assessed by bootstrap method[48] with 100 replicates.

**Modeling studies.** The AtCHIL crystal structure (PDB ID: 4DOK) was obtained from the Protein Data Bank (http://www.rcsb.org/pdb/) as the receptor molecule. The known structure of *p*-coumaroyl-diketide-CoA (PubChem CID: 70679033) was used to determine the structure of *p*-coumaroyl-tetraketide-CoA. Discovery Studio[49] and Schrödinger (Schrödinger, LCC, New York, NY, USA) suites were used for modeling, structure creation, and validation. MAESTRO/SiteMap[50] was used for identifying the binding sites on the surface of the AtCHIL structure, whereas GLIDE[51] and GOLD[52] were used to analyze docking. For all docked complexes, we calculated the Prime/Molecular Mechanics Generalized Born Surface Area function to estimate the Gibbs energy of binding ($\Delta G$ of binding) (Prime/MM-GBSA[53]) and the Piecewise Linear Potential (chemPLP) function to estimate the accuracy of the docking[54], with the hydrogen bonding term and multiple linear potentials used to model van der Waals and repulsive terms.

**Reporting summary.** Further information on research design is available in the Nature Research Reporting Summary linked to this article.

## Data availability

All data needed to evaluate the conclusions in the manuscript are presented herein and/or as Supplementary Information. Additional data related to this study may be requested from the authors. The sequences for all genes described in this manuscript are available in the GenBank/EMBL/DDBJ databases and other databases shown in Supplementary Table 4. The source data underlying Figs. 4–6 as well as the source data of the gels and immunoblots in Supplementary Figs. 4, 6b, and 8 are provided in the Source Data file.

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

## Acknowledgements

We are grateful to Prof. Richard L. Smith, Graduate School of Environmental Studies, Tohoku University, for his invaluable advice. We thank Edanz Group (www.edanz-diting.com/ac) for editing a draft of this manuscript. This study was supported in part by a JSPS KAKENHI grant (18H03938) and the NIBB Collaborative Research Program (14-388, 15-352, 16-319, 17-332, and 18-337).

## Author contributions

T.W., K.S., K.Y.-S., Y.M., A.H., S.T. and T. Nakayama conceived and designed experiments; T.W., R.M., T. Nakano, S. Yamada, S. Yamashita, M.T., K.I., N.T., Y.L., N.F. K.U., Y.A., S.S. and K.Y.-S. performed the experiments; T.W., K.Y.-S., Y.M. and A.H. contributed plant materials, reagents, and analytical tools; K.D. performed the computational docking studies; Y.K., T.W. and T. Nakayama performed the phylogenetic analyses; T.W. and T. Nakayama wrote the paper.

## Competing interests

The authors declare no competing interests.

**Additional information**

