## [Peer Review File · Nature Communications]

Reviewers' comments:

Reviewer #1 (Remarks to the Author):

The manuscript entitled "Rectifying the catalytic promiscuity: A conserved strategy of chalcone isomerase-like protein to correct chalcone synthase specificity" presents very interesting finding of the functional role of chalcone isomerase-like proteins (CHILs). It has been shown that CHI has evolved from a fatty acid-binding protein (FABP), but there is intermediate group (CHILs) between CHI and FABP in the phylogeny, and the function of CHILs (they are clearly neither CHI nor FABP_ are mystery for a long time.

In this paper, the authors beautifully demonstrated that CHILs are bind to CHS to enhance THC production and decrease CTAL formation, and rectifying the promiscuous CHS catalysis. The authors presented a large bulk of experimental data to convincingly show that CHIL binds to CHS and act as rectifier to reduce the promiscuous CHS catalysis. Such function of proteins is rather unusual and would be of a great interest in the community of enzymology/biochemistry in general. I believe that the work presents highly important findings and suitable to publish in Nature Communications.

I have only one problem in this paper.

While I do not have any problem with all biochemical experimental data the authors presented. I have to say that the phylogenetic analysis they presented does not meet a standard to publish in a good journal. First, they have employed neighbor joining method to infer the phylogenetic tree, which is the most unreliable method among all other methods (it is simply quick and dirty method, and the common sense is that it is not a good method). I suggest rerun the phylogeny with more reliable and standard methods such as Maximum Likelihood or Maximum Parsimony (ideally few different methods), for example, IQ-tree (<http://www.iqtree.org/>) provides a nice interface for an non-expert to generate much more reliable tree for publication. I also cannot understand what the authors discuss what they found in the phylogenetic relationships from the tree. I suspect that they will see much different tree from what they present (so I would not comment much in the current text), but they should pub clear explanations about the phylogeny and what they discuss in the text (more clear figure caption).

Reviewer #2 (Remarks to the Author):

This manuscript reports exciting results. CHIL is known to stimulate flavonoid formation. Here the authors not only detect the physical interaction between CHS and CHIL in the flavonoid metabolon but also demonstrate that this interaction is PKS- and species-specific. Most importantly, CHIL greatly increases the formation of THC at the expense of CTAL. Rectifying the promiscuity of an enzyme via interaction with components of the metabolon is likely to apply to other pathways in plant specialised metabolism, which will stimulate research into this interesting field. The authors have used an array of techniques and have refined procedures employed. Before consideration for publication, the following aspects may be revised. CHIL sequences were amplified from a number of species. If the protein is encoded by a gene family, isoforms may differ in their interaction and rectifying potential. A 5- to 10-fold molar excess of CHIL over CHS was found to be highly efficient. Can the authors estimate the ratio present in the engineered E. coli strain, e.g. antibody-based? Co-expression may lead to an improved interaction compared to simple mixing and hence result in higher, possibly exclusive, THC production, provided the excess of CHIL is ensured. In this context, how to rule out that the observed 100 kDa band refers to two AtCHIL monomers on one AtCHS monomer? Previously, CHI was found to bind to CHIL. Given the small size of CHIL, the ratio of the CHS products may be affected by additional binding of CHI, which could be tested by co-expression of CHI in the engineered E. coli strain. In the co-precipitation experiment, 150 and 192 proteins were able to specifically bind to the His6-target proteins. How specific is this binding, given the high numbers of proteins? After clarification of these aspects, a publication in Nature

Communications can be taken into consideration. Minor points: line 64, please correct 'hydroxycinnamoyl'; line 65, please add 'general' phenylpropanoid.

Responses to Reviewers

(Reviewers comments are shown with a blue font)

Reviewer #1:

Comments:

The manuscript entitled “Rectifying the catalytic promiscuity: A conserved strategy of chalcone isomerase-like protein to correct chalcone synthase specificity” presents very interesting finding of the functional role of chalcone isomerase-like proteins (CHILs). It has been shown that CHI has evolved from a fatty acid-binding protein (FABP), but there is intermediate group (CHILs) between CHI and FABP in the phylogeny, and the function of CHILs (they are clearly neither CHI nor FABP_ are mystery for a long time.

In this paper, the authors beautifully demonstrated that CHILs are bind to CHS to enhance THC production and decrease CTAL formation, and rectifying the promiscuous CHS catalysis. The authors presented a large bulk of experimental data to convincingly show that CHIL binds to CHS and act as rectifier to reduce the promiscuous CHS catalysis. Such function of proteins is rather unusual and would be of a great interest in the community of enzymology/biochemistry in general. I believe that the work presents highly important findings and suitable to publish in Nature Communications.

I have only one problem in this paper. While I do not have any problem with all biochemical experimental data the authors presented. I have to say that the phylogenetic analysis they presented does not meet a standard to publish in a good journal. First, they have employed neighbor joining method to infer the phylogenetic tree, which is the most unreliable method among all other methods (it is simply quick and dirty method, and the common sense is that it is not a good method). I suggest rerun the phylogeny with more reliable and standard methods such as Maximum Likelihood or Maximum Parsimony (ideally few different methods), for example, IQ-tree (<http://www.iqtree.org/>) provides a nice interface for an non-expert to generate much more reliable tree for publication. I also cannot understand what the authors discuss what they found in the phylogenetic relationships from the tree. I suspect that they will see much different tree from what they present (so I would not comment much in the current text), but they should pub clear explanations about the phylogeny and what they discuss in the text (more clear figure caption).

Reply:

Thank you very much for your constructive comments on our paper and very important suggestions on our phylogenetic analysis and presentation of the results. In accordance with your suggestion, we have rerun the phylogenetic

analysis with the recommended method (maximum likelihood [ML] method), as described in the Methods section of the revised manuscript (p. 21, lines 663–678). The results obtained were compared with those obtained in the original manuscript (using the neighbor-joining [NJ] method). Phylogenetic trees obtained by each method are presented in Supplementary Fig. 10 (c)/(d) and (a)/(b), respectively, of the revised manuscript. Analyses by both methods yielded very similar tree topologies. Therefore, the conclusion derived from the phylogenetic analyses was not changed from that described in the original manuscript. However, consistent with your suggestion, we carefully rewrote the discussion to make the point more clearly (p. 12, lines 365–387) and included additional detail in the figure captions (Supplementary Materials p. 17; Supplementary Fig. 10). Thank you very much again for your valuable suggestions for improvement of our paper.

Reviewer #2 (Remarks to the Author):

Comments 1:

This manuscript reports exciting results. CHIL is known to stimulate flavonoid formation. Here the authors not only detect the physical interaction between CHS and CHIL in the flavonoid metabolon but also demonstrate that this interaction is PKS- and species-specific. Most importantly, CHIL greatly increases the formation of THC at the expense of CTAL. Rectifying the promiscuity of an enzyme via interaction with components of the metabolon is likely to apply to other pathways in plant specialised metabolism, which will stimulate research into this interesting field. The authors have used an array of techniques and have refined procedures employed. Before consideration for publication, the following aspects may be revised.

Reply:

Thank you very much for your constructive comments on our paper and very important suggestions. To address your comments, we have performed additional experiments and carefully revised the manuscript.

Comments 2:

CHIL sequences were amplified from a number of species. If the protein is encoded by a gene family, isoforms may differ in their interaction and rectifying potential.

Reply:

Plant species, including those used in our study, generally encode only a single CHIL gene in their genomes. The exceptions include soybean, which encodes two isoforms of CHIL in its genome. However, our transcriptome data showed that only one of the two CHIL isoforms of soybean was highly expressed, therefore we did not characterize the isoform that showed an extremely low

expression level.

We agree with the reviewer's comment that different CHIL isoforms, if any, may potentially differ in their interaction and rectifying potential if the primary structures of the CHIL isoforms differ with each other. The validity of this idea can be implicated from the results shown in Fig. 3, panel (b) and Fig. 4. The results show that InCHL (although not an isoform of AtCHIL) could weakly bind to AtCHS (Fig. 3, panel (b)) and partly fulfill the role as an enhancer of flavonoid production (Fig. 4).

Comments 3:

A 5- to 10-fold molar excess of CHIL over CHS was found to be highly efficient. Can the authors estimate the ratio present in the engineered *E. coli* strain, e.g. antibody-based? Co-expression may lead to an improved interaction compared to simple mixing and hence result in higher, possibly exclusive, THC production, provided the excess of CHIL is ensured.

Reply:

Thank you very much for your important comment. We believe that by performing experiments using *in vivo E. coli* systems, we were able to at least partly show the CHIL-mediated enhancement of chalconoid (CLC) ratio in the total CHS products. In accordance with your suggestion, we estimated the molar ratios of CHIL:CHS present in the engineered *E. coli* strains by means of immunoblot analysis. The results showed that the molar ratios (CHIL:CHS) were 1.5:1.0 in strain (+). These new results are included in the text (p. 8, lines 247–249) and Supplementary Fig. 8 of the revised manuscript.

We agree with the reviewer's idea that co-expression may lead to an improved interaction compared with simple mixing. As described in our reply to your Comment 5, this issue could be addressed in conjunction with protein–protein interactions in a metabolon formed in the intracellular milieu in our future studies.

Comments 4:

In this context, how to rule out that the observed 100 kDa band refers to two AtCHIL monomers on one AtCHS monomer?

Reply:

Thank you very much for your important comment. To address your comments, we repeated similar experiments using different concentrations of CHS, CHIL, and BS3 (a cross linker) using the snapdragon system. The results showed that the 70-kDa band was the only protein band that was specifically produced in the presence of the cross linker. The 100-kDa band, which we mentioned in the

original manuscript, was not reproducibly observed. These results are presented in Supplementary Fig. 6 and described on p. 5, lines 145–155 of the revised manuscript.

Comments 5:

Previously, CHI was found to bind to CHIL. Given the small size of CHIL, the ratio of the CHS products may be affected by additional binding of CHI, which could be tested by co-expression of CHI in the engineered *E. coli* strain.

Reply:

We thank the reviewer for this comment. The engineered *E. coli* strain expresses 4CL, CHS, and CHIL from soybean (Gm4CL-3, GmCHS-1, and GmCHIL). Our results showed that soybean CHI isozymes (GmCHI-1A, GmCHI-1B2, and GmCHI-2) did not interact with soybean CHIL (GmCHIL; Fig. 2 (b), *right panels*). Moreover, our previous analyses showed the absence of physical interaction between these soybean CHI isozymes and soybean CHS isozymes (GmCHS-1 and GmCHS-7) (Waki, T. *et al. Biochem. Biophys. Res. Commun.*, **469**, 546-551 (2016); Mameda, R., *et al., Plant J.*, **96**, 56-74 (2018)). Therefore, we did not examine the effect of GmCHI co-expression in the engineered *E. coli* strain.

However, we acknowledge that the reviewer's comment points to a very important issue to be addressed in our future studies. Specifically, our recent studies showed that CHS, CHI, CHIL, and some other flavonoid enzymes form a metabolon on the endoplasmic reticulum, where a cytochrome P450 monooxygenase, such as flavone synthase II and isoflavone synthase, serves as a nucleus of metabolon formation. Thus, it would be tempting to speculate that the e-values may be further increased by the involvement of CHIL in the flavonoid metabolon. This could be tested by co-expression of CHIL, flavonoid enzymes, together with membrane-bound P450 in the engineered yeast strain in our future studies. We thank the reviewer again for the important comment.

Comments 6:

In the co-precipitation experiment, 150 and 192 proteins were able to specifically bind to the His6-target proteins. How specific is this binding, given the high numbers of proteins?

Reply:

The co-precipitation experiments were carefully performed with appropriate controls, and the high numbers (150 and 192) of "specifically bound" proteins were identified via LC/MS/MS analysis. We admit that additional lines of evidence would be needed to conclusively show that each of these 150 and 192

proteins specifically bind to AmCHIL and AmCHS, respectively, with physiological significance. The primary objective of the co-precipitation experiments was to confirm that the CHS–CHIL interaction indeed takes place in a native plant system (snapdragon, in this case), because the conclusion concerning protein–protein interactions between CHS and CHIL were mainly based on their detections using heterologous interaction assay systems (i.e., a yeast two-hybrid system and BiFC with a *Nicotiana benthamiana* system). The fact that CHS and CHIL were detected among proteins bound to CHIL and CHS, respectively, in the snapdragon flower extracts was at least supportive of the protein–protein interaction between CHS and CHIL in the native plant (snapdragon) system. In the revised manuscript, we added the data in which the CHS–CHIL interaction was detected via an immunoblot analysis after co-precipitation (p.5, lines 129-131; and Supplementary Fig. 4). Collectively, both of these results are consistent with the notion that the CHS–CHIL interaction indeed takes place in the native plant (snapdragon) system.

Comments 7:

After clarification of these aspects, a publication in *Nature Communications* can be taken into consideration.

Minor points:

line 64, please correct 'hydroxycinnamoyl'

line 65, please add 'general' phenylpropanoid.

Reply:

We are grateful to the reviewer for suggesting these revisions. Each of the errors have been corrected as suggested (p. 3, lines 64 and 65).

REVIEWERS' COMMENTS:

Reviewer #1 (Remarks to the Author):

I am satisfied the revision and I believe that the manuscript is good to publish in Nature Communications.

Reviewer #2 (Remarks to the Author):

The authors have well responded to the criticism raised. They have carried out additional experiments and have properly changed the manuscript.

AUTHORS' REPLY TO REVIEWERS' COMMENTS

Reviewers' Remarks to the Author

Reviewer #1:

I am satisfied the revision and I believe that the manuscript is good to publish in Nature Communications.

Reviewer #2:

The authors have well responded to the criticism raised. They have carried out additional experiments and have properly changed the manuscript.

Authors' Reply

We deeply thank the reviewers again for their constructive comments and invaluable suggestions to improve this paper.